# Learning 3D Particle-based Simulators from RGB-D Videos

**William F. Whitney,**[*] **Tatiana Lopez-Guevara,**[*] **Tobias Pfaff, Yulia Rubanova,**
**Thomas Kipf, Kimberly Stachenfeld, Kelsey R. Allen**
Google DeepMind

## Abstract

Realistic simulation is critical for applications ranging from robotics to animation. Traditional analytic simulators sometimes struggle to capture sufficiently realistic simulation which can lead to problems including the well known "sim-to-real" gap in robotics. Learned simulators have emerged as an alternative for better capturing real-world physical dynamics, but require access to privileged ground truth physics information such as precise object geometry or particle tracks. Here we propose a method for learning simulators directly from observations. Visual Particle Dynamics (VPD) jointly learns a latent particle-based representation of 3D scenes, a neural simulator of the latent particle dynamics, and a renderer that can produce images of the scene from arbitrary views. VPD learns end to end from posed RGB-D videos and does not require access to privileged information. Unlike existing 2D video prediction models, we show that VPD's 3D structure enables scene editing and long-term predictions. These results pave the way for downstream applications ranging from video editing to robotic planning.

## 1 Introduction

Physical simulation underpins a diverse set of fields including robotics, mechanical engineering, game development, and animation. In each of these fields realistic simulation is critical and substantial effort goes into developing simulation engines that are specialized for the unique requirements and types of physical dynamics of that field (Todorov et al., 2012; OpenCFD, 2021; Coumans & Bai, 2016). Obtaining sufficiently realistic simulators for a particular scene requires additional work (Shao et al., 2021), including the tuning of simulation assets like object models and textures, as well as the physical properties that will lead to realistic dynamics (Qiao et al., 2022). Even with significant effort and tuning, it is often impossible to perfectly capture the physics of any particular real-world scene, as sub-scale variations in surfaces and textures can have significant impacts on how objects behave.

Over the last few years, learned simulators have emerged as an alternative to carefully hand-crafted analytic simulators. They can be trained to correct the outputs of analytic solvers (Kloss et al., 2022; Ajay et al., 2018), or trained to mimic analytic physics directly, but at significantly faster speeds (Pfaff et al., 2021; Kochkov et al., 2021). Learned simulators can capture many different dynamics, ranging from liquids and soft materials to articulated and rigid body dynamics (Li et al., 2019; Allen et al., 2023). When researchers can provide state-based information for a real-world dynamic scene (the positions and poses of objects over time), learned simulators can mimic real-world physics better than careful tuning with analytic simulators (Allen et al., 2022).

However, learned simulators still require access to privileged "ground truth" physics information to be trained. The vast majority require near-perfect state estimation – the poses, exact 3D shapes, and positions – of all objects, at all time points, in a scene. Recent attempts to relax these requirements still require access to other forms of supervised information such as object segmentation masks (Driess et al., 2022; Riochet et al., 2020; Shi et al., 2022).

While there exist many video prediction models that do not require such privileged information (Finn et al., 2016; Clark et al., 2019; Ho et al., 2022), these models are not simulators. Unlike simulators, 2D video models do not operate in 3D, do not generally support rendering a dynamic scene from a

---

[*]Equal contribution. Correspondence to {`wwhitney,zepolitat`}@google.com.

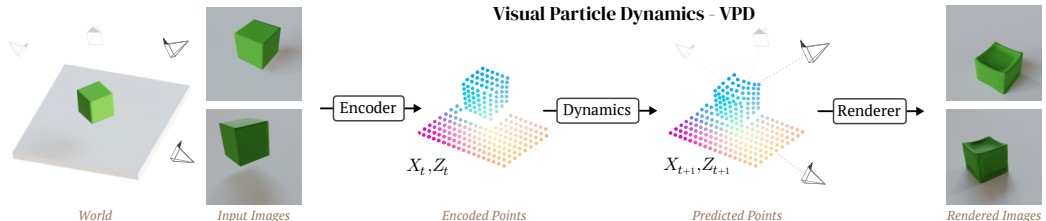

Figure 1: A diagram of our model. The encoder maps multiple RGB-D cameras into a latent point cloud representation using a convolutional encoder. The dynamics model, a hierarchical graph neural network, learns to transform this point cloud to a new point cloud on the next time-step. The 3D latent point cloud is decoded into an image, supporting an image-based loss that can be used to train the entire pipeline end-to-end.

new viewpoint, and do not allow 3D editing of scenes to produce new simulation outcomes. Recent works such as NeRF-dy Li et al. (2021a) which represent scenes as single latent vectors can support rendering from new viewpoints, but cannot support 3D editing. Similarly, recent advances in 3D neural representations for dynamic scenes (Li et al., 2023; Du et al., 2021c), can provide beautiful reconstructions of single videos without access to privileged physics information, but these models are not simulators either. Once trained, they represent a recording of a specific scene, and cannot be applied to a new scene without re-training.

Here we ask whether we can learn *simulators* from multi-view RGB-D observations alone. Our model, Visual Particle Dynamics (VPD), jointly learns a latent particle-based 3D scene representation, a predictive model represented as a hierarchical graph neural network on that representation, and a conditional renderer that can produce new images from arbitrary views. It is trained end to end on multi-view RGB-D data without object masks or 2D segmentation models, and has the following desirable properties which we demonstrate:

- VPD supports 3D state editing. Its explicit 3D representation can be edited, simulated, and re-rendered from novel views.

- VPD supports multi-material simulation. We demonstrate results for multi-body rigid dynamics and soft-body interactions.

- VPD outperforms 2D video models in data efficiency. With as few as 16 trajectories, VPD can learn a simulator for a simple dynamic scene.

To our knowledge, Visual Particle Dynamics is the first fully learned simulator that supports these crucial properties, and which does not require access to any privileged supervised information for training or inference.

## 2 RELATED WORK

Given the importance of realistic simulation across a wide variety of subject areas (robotics, graphics, engineering), there are a variety of techniques for learning predictive models with different capabilities, each with different assumptions about input data requirements. We outline the general classes of learned predictive models below, and summarize the capabilities of key related works in Table 1.

**Learned simulators** Learned simulators aim at replacing analytic simulators with a learned function approximator. This means they are generally trained on *state* information from a ground truth solver, and often cannot be used directly on visual input data. Depending on the application, state can be represented as point clouds (Li et al., 2019; Sanchez-Gonzalez et al., 2020; Mrowca et al., 2018), meshes (Pfaff et al., 2021; Allen et al., 2023), or as SDFs (Le Cleac'h et al., 2023). Learned function approximators such as graph neural networks (GNNs) (Battaglia et al., 2018), continuous convolutional kernels (Ummenhofer et al., 2019), or MLPs (Li et al., 2021a), can then be used to model the evolution of the state through time. Our work goes beyond these approaches by never requiring access to the states directly, instead learning from RGB-D videos.

| Method | Supervision | | Sensors | Capabilities | | | |
|---|---|---|---|---|---|---|---|
| | No states | No masks | Type | Editable | Rendering | 3D | Prediction |
| NeRF-dy (Li et al., 2021a) | ✓ | ✓ | multi RGB | ✗ | ✓ | ✗ | ✓ |
| 3D-IntPhys (Xue et al., 2024) | ✗ | ✗ | multi RGB | ✓ | ✓ | ✓ | ✓ |
| RoboCraft (Shi et al., 2022) | ✓ | ✗ | multi RGB-D | ✓ | ✗ | ✓ | ✓ |
| SlotFormer (Wu et al., 2023) | ✓ | ✓ | RGB | ✗ | ✓ | ✗ | ✓ |
| Driess et al. (2022) | ✓ | ✗ | multi RGB | ✓ | ✓ | ✓ | ✓ |
| Dynibar (Li et al., 2023) | ✓ | ✓ | multi RGB | ✗ | ✓ | ✓ | ✗ |
| Ours | ✓ | ✓ | multi RGB-D | ✓ | ✓ | ✓ | ✓ |

Table 1: Comparison of input data requirements and capabilities of prior methods. VPD learns a full dynamics *prediction* model with *3D* bias, *editability*, and free-camera *rendering*, without requiring *state* information or segmentation *masks*, but does require multiple RGB-D sensors.

**Bridging perception and simulators** Several approaches rely on these learned simulators as a dynamics backbone, but attempt to learn perceptual front-ends that can provide point clouds or mesh information from video. Some attempts rely on *pre-trained* dynamics models which were trained with ground truth state information (Guan et al., 2022; Wu et al., 2017), and learn a mapping from perception to 3D state. Learning this mapping for a general setting is a very hard task, and hence most methods only work for simple scenes, and/or require object segmentation masks to seperate the scene into components (Janner et al., 2019; Driess et al., 2022; Shi et al., 2022; Xue et al., 2024). In contrast, VPD removes the requirement of either having a pre-trained simulator or object segmentation masks and trains end to end with pixel supervision. NeRF-dy (Li et al., 2021a) reverses this formula, first learning a vector representation of scenes, then learning a dynamics model on this representation. We describe NeRF-dy in Section 4.1 and compare to its choices in Section 5.

**Analytic simulators and system identification** Another strategy for constructing a simulation of a real-world scene involves modeling the scene in an analytic simulator, then fitting the physics coefficients of this simulator to match the real data as well as possible. Differentiable simulators (Hu et al., 2020b; Freeman et al., 2021; Du et al., 2021a; Howell et al., 2022; Macklin, 2022) enable the use of gradients in system identification; however, this typically requires ground-truth object positions. Recent work (Qiao et al., 2022; Le Cleac'h et al., 2023) combines implicit 3D object representations with a differentiable simulator to support capturing object models from the real world. Methods for inferring articulated models from video (Heiden et al., 2022) similarly depend on pre-trained instance segmentation models and have only been applied to simple settings. Overall these methods can give good results, but are limited by the types of physics representable in the analytic simulator and the complexities of state estimation.

**Unsupervised video models** Video models do not require object segmentation masks or pre-trained dynamics models. Some methods directly model dynamics in pixel-space (Finn et al., 2016; Clark et al., 2019; Ho et al., 2022), but often fail to model dynamics reliably. Other models attempt to discover object representations (Du et al., 2021b), or more recently, simultaneously learn object representations and dynamics (Watters et al., 2017; Qi et al., 2020; Wu et al., 2023). Given the breadth of existing video data, large-scale learning with transformers and diffusion models has shown promise for enabling long-horizon planning in robotics (Hafner et al., 2023; Du et al., 2023). However, these learned models are not simulators. Those that learn dynamics do not generally operate in 3D and they do not support compositional editing. VPD possesses both of these qualities while still learning from fully unsupervised data.

**Video radiance fields** A class of methods extend NeRF (Mildenhall et al., 2020) with a temporal axis, to encode individual videos in neural network weights (Li et al., 2021b; Du et al., 2021c; Li et al., 2023). Unlike the methods above however, these approaches encode a single video trajectory and do not support predicting videos, or generalizing across different videos. In contrast, VPD supports both.

## 3 VISUAL PARTICLE DYNAMICS

A Visual Particle Dynamics (VPD) model consists of three components: (1) an *encoder* which maps a set of one or more RGB-D images into a 3D latent particle representation; (2) a *dynamics model*

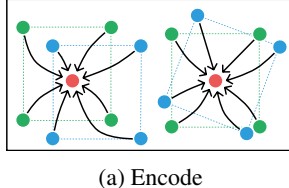
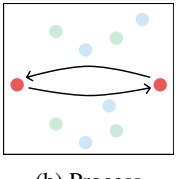
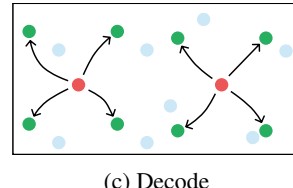

| (a) Encode | (b) Process | (c) Decode |

Figure 2: Hierarchical message passing. **(a)** The particle representing the past timestep $t - 1$ (blue), and particles representing the present state $t$ (green) send their features and relative positions to the abstract nodes (red). **(b)** The abstract nodes perform multi-step message passing amongst themselves. **(c)** The abstract nodes send updates back to the present-time particle nodes.

which predicts the evolution of these particles through time; and (3) a *renderer* which decodes the particles into an image. See Figure 1 for an overview of our method. All of these components are trained together and end to end via supervision on a multi-step pixel-wise loss.

## 3.1 ENCODER

The encoder converts a set of RGB-D images, corresponding to a short video captured from one or more views, into a set of latent particles for each time-step. First, we unproject each pixel into 3D space using each camera's transformation matrix and per-pixel depth value to obtain the 3D location $\boldsymbol{X}_{ij}$. Next, we process the input RGB image using a UNet (Ronneberger et al., 2015) to produce per-pixel latent features $\boldsymbol{Z}_{ij}$. Latent features and 3D locations for each pixel $(i, j)$ are combined to form a latent particle $(\boldsymbol{X}_{ij}, \boldsymbol{Z}_{ij})$ as shown in Figure 7.

This procedure produces a set of particles $\mathcal{P}$ for each image. If more than one camera is present, we can simply merge all the particle sets from each time-step. The UNet shares its weights between all cameras and does not receive camera or depth inputs. In our experiments, we filter the set of particles to those that lie within a designated area containing the objects of interest, which we refer to as the workspace (see Table 3). We then subsample the particles uniformly at random to limit the memory footprint of the network. See Appendix D for results with varying numbers of particles.

## 3.2 DYNAMICS MODEL

The dynamics model predicts the set of particles at time $t + 1$ given the current and previous particles using a GNN, i.e. $\hat{\mathcal{P}}^{t+1} = D(\mathcal{P}^{t-1}, \mathcal{P}^t)$, and is applied recursively to make multi-step particle rollouts. Prior work on GNN-based dynamics models generally assumes ground truth correspondences between points at different time-steps (Sanchez-Gonzalez et al., 2020), and use those to compute e.g. velocities. However, correspondences are not available when working with video, and can be unreliable or unspecified (e.g. objects moving out of frame). We therefore decide to entirely side-step finding correspondences. Instead, we connect latent points of different time-steps in a graph based on spatial proximity and rely on the network to learn to correctly propagate information.

**Graph construction** The most straightforward approach to construct such a graph would be to connect each point in $\mathcal{P}^t$ to all points in $\mathcal{P}^{t-1}$ within a certain spatial distance. However, working with videos requires a large number of points ($\sim 2^{14}$) to accurately represent the textures and geometry of a complex scene. Directly connecting these point clouds would generate an excessive number of graph edges. Instead, we employ a hierarchical 2-layer GNN (Cangea et al., 2018; Ying et al., 2018; Lee et al., 2019) to sparsify connections. Inspired by Hu et al. (2020a); Fortunato et al. (2022), we construct a sparse layer of "abstract nodes" $\mathcal{A}$, whose locations are chosen uniformly at random from $\mathcal{P}^t$, thereby roughly matching its point density. Each point in $\mathcal{P}^{t-1}, \mathcal{P}^t$ connects to up to two nearest abstract nodes within a small spatial radius $r_s$, enabling the abstract nodes to convey information over time. The abstract nodes are connected with bidirectional edges to other abstract nodes within a larger radius $r_s^a$, as well as reciprocal edges to each connected node in $\mathcal{P}^t$ (Figure 2). Communicating through this sparse set of abstract nodes significantly reduces the number of necessary edges. Details on graph construction and parameters can be found in Appendix A.

**Message passing** To compute the dynamics update, we use multigraph message passing with an Encode-Process-Decode architecture (Pfaff et al., 2021). The multigraph $\mathcal{G}$ is defined by the particle

nodes $\mathcal{V}_{\mathcal{P}^{t-1}}, \mathcal{V}_{\mathcal{P}^t}$, abstract nodes $\mathcal{V}_{\mathcal{A}}$, and edges $\mathcal{E}_{\mathcal{P}^{t-1} \to \mathcal{A}}, \mathcal{E}_{\mathcal{P}^t \to \mathcal{A}}, \mathcal{E}_{\mathcal{A} \to \mathcal{A}}, \mathcal{E}_{\mathcal{A} \to \mathcal{P}^t}$ between these node sets. We encode latent features $\mathbf{Z}$ from the image encoder into the nodes $\mathcal{V}_{\mathcal{P}}$, and the distance vector between node positions into the edges. Nodes $\mathcal{V}_{\mathcal{A}}$ do not receive any initial features.

Next, we perform one round of message passing on all nodes $\mathcal{V}_{\mathcal{P}}$ to abstract nodes $\mathcal{V}_{\mathcal{A}}$, followed by $K = 10$ message passing steps between abstract nodes, and one message passing step back from $\mathcal{V}_{\mathcal{A}}$ to $\mathcal{V}_{\mathcal{P}^t}$. Finally, the features in $\mathcal{V}_{\mathcal{P}^t}$ are decoded to the difference vectors $(\Delta \hat{\mathbf{x}}, \Delta \hat{\mathbf{z}})$ in position, and latent state. These vectors are added to the particle state $\mathcal{P}^t$ to form the estimate $\hat{\mathcal{P}}^{t+1}$ used for rendering. Since the renderer operates on relative locations, the predicted $\Delta \hat{\mathbf{x}}$ directly corresponds to objects moving in the predicted video, while changes in $\Delta \hat{\mathbf{z}}$ allows for predicting appearance changes caused by motion, such as a shadow moving beneath a flying object.

Algorithm 1 in Appendix A details the entire message passing algorithm. Following Pfaff et al. (2021), we use MLPs for encoding, decoding and message passing, with separate sets of weights for different edge and node types. See Appendix A for all architectural details.

### 3.3 RENDERER

We use a neural renderer similar to (Xu et al., 2022; Guan et al., 2022) to render a particle set into an image from a desired camera. Following the volumetric rendering formulation from Mildenhall et al. (2020), rendering a given ray consists of querying a neural network at a set of locations along the ray to get an RGB color and a density, then compositing these values along the ray to get the color of a single pixel. The renderer $R$ predicts a color $\hat{\mathbf{c}}$ given a particle set and the origin and direction $(\mathbf{o}, \mathbf{d})$ of this ray: $\hat{\mathbf{c}} = R(\mathcal{P}, (\mathbf{o}, \mathbf{d}))$. For VPD, and similar to Xu et al. (2022), the input features at each query location are computed from a set of $k$ approximate nearest neighbor particles (see Appendix I).

Let $p_x$ be the location of a particle and $p_z$ be its latent features, with $\mathcal{N}(\mathbf{x})$ the set of particles close to a location $\mathbf{x}$. We compute the features at a location $\mathbf{x}$ using a kernel $k$ to be $f_k(\mathbf{x}) = \sum_{p^i \in \mathcal{N}(\mathbf{x})} k(\mathbf{x}, p_x^i) \cdot p_z^i$. For VPD, we use a set of concentric annular kernels defined by a radius $r$ and a bandwidth $b$:

$$k_{r,b}(\mathbf{x}, \mathbf{x}') = \exp \left\{ -\frac{(\|\mathbf{x}' - \mathbf{x}\|_2 - r)^2}{b^2} \right\} \tag{1}$$

This kernel weights points more highly when they are at a distance $r$ from the query location. We concatenate the computed features using a set of $m$ kernels $k_1 \dots k_m$, giving the final feature vector

$$f(\mathbf{x}) = f_{k_1}(\mathbf{x}) \oplus \dots \oplus f_{k_m}(\mathbf{x}) \tag{2}$$

where $\oplus$ denotes concatenation. The network then predicts the color $\hat{\mathbf{c}}$ and density $\hat{\sigma}$ at a location $\mathbf{x}$ and a viewing direction $\mathbf{u}$ as $(\hat{\mathbf{c}}, \hat{\sigma}) = \text{MLP}(f(\mathbf{x}) \oplus \gamma(\mathbf{u}))$, where $\gamma$ is NeRF positional encoding.

### 3.4 TRAINING

Given a multi-view RGB-D video, we train VPD by encoding the first two timesteps into particle sets $(\mathcal{P}^1, \mathcal{P}^2)$. We then recursively apply the dynamics model $\hat{\mathcal{P}}^{t+1} = D(\mathcal{P}^{t-1}, \mathcal{P}^t)$ to generate particle sets $(\hat{\mathcal{P}}^3, \dots, \hat{\mathcal{P}}^{2+T})$ for some prediction length $T$. The complete loss is the expected squared pixel reconstruction error over timesteps $t = 3 \dots 2 + T$ and ray indices $i$ across all cameras:

$$\mathcal{L}_{\text{VPD}} = \frac{1}{T} \sum_{t=3}^{2+T} \mathbb{E}_i \left[ R\left(\hat{\mathcal{P}}^t, (\mathbf{o}_i^t, \mathbf{d}_i^t)\right) - \mathbf{c}_i^t \right]^2 \tag{3}$$

In our training we use 256 sampled rays per timestep to evaluate this expectation.

## 4 EXPERIMENTAL SETUP

We test VPD on three datasets which stress different simulator capabilities. The MuJoCo block dataset (Todorov et al., 2012) is visually simple but tests a model's ability to accurately represent crisp rigid contact (Allen et al., 2022). The Kubric datasets (Greff et al., 2022) encompass a range of visual complexities, from Platonic solids to densely-textured scans of real objects and backgrounds,

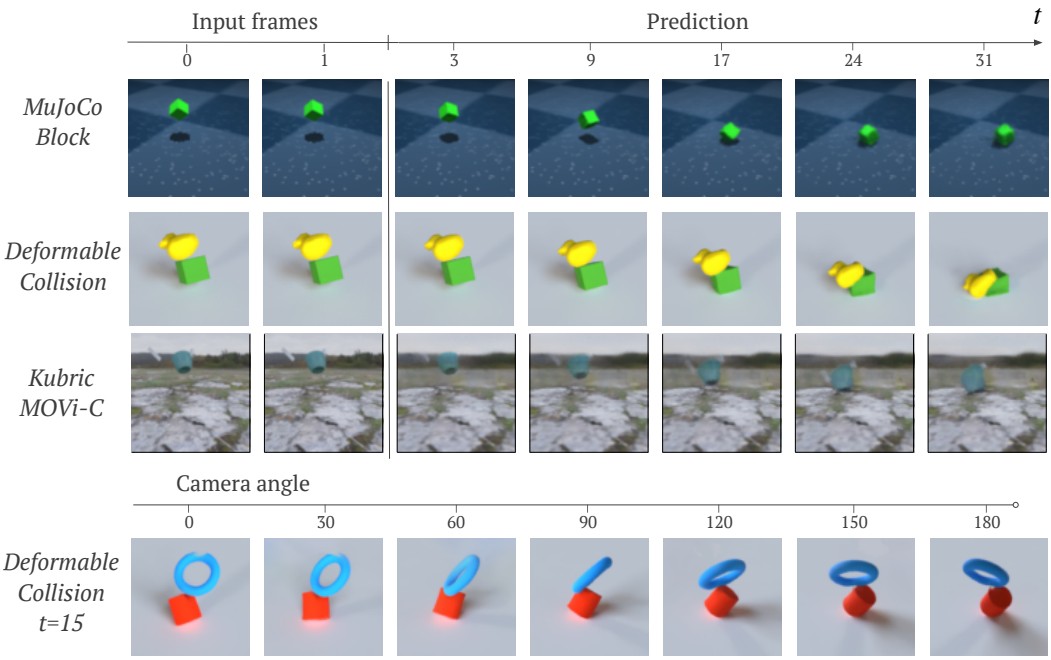

Figure 3: **Top**: Example rollouts from VPD on different datasets. **Bottom**: Viewpoint generalization; VPD is unrolled for 15 time steps, and the latent state is rendered from novel camera positions. See our video site and Appendix F, Appendix G for more rollouts.

and tests a model's ability to represent multi-object interactions in varied visual environments. The deformable dataset evaluates a model's ability to represent the dynamics of non-rigid objects with a large number of degrees of freedom. In all cases, the models are provided with RGB-D views from multiple cameras. For evaluation, 16 trajectories are chosen at random and held out from each dataset, and we report each model's PSNR (with SSIM in Appendix E) (Wang et al., 2004). The PSNR (Peak Signal to Noise Ratio) correlates with the mean squared error between the ground truth and predicted images, and therefore captures both deviations in the dynamics and effects of blurring.

**MuJoCo block** contains 256 trajectories, each with 64 time-steps, of a single block being tossed onto a plane with randomized initial height, velocity, and angular velocity. RGB-D images are generated from 16 $128 \times 128$ pixel resolution cameras arranged in a half sphere.

**Kubric Movi-A/B/C** (Greff et al., 2022) includes trajectories of 3 objects (from a set of 3, 10, and 1033 different possible shapes, respectively) being thrown onto a grey/coloured/textured floor with randomized initial velocities and sizes. We re-generate the dataset to include camera information. RGB-D images are rendered from 9 cameras of $128 \times 128$ pixel resolution arranged in a half dome. 350 trajectories are used for training, each with 96 time-steps.

**Deformables** is a dataset of deformable objects, simulated and rendered using Blender (Blender, 2018) softbody physics. In Deformable Block a cube is dropped onto the floor with randomized initial position and orientation. In Deformable Multi, one object out of a set of five (Cube, Cylinder, Donut, Icosahedron, Rubber duck) is dropped onto the floor with randomized initial position and orientation. In Deformable Collision two objects are dropped and collide onto the floor. All datasets consist of 256 training trajectories with 80 time steps. The scene is rendered using four cameras of $128 \times 128$ pixel resolution arranged in a half dome.

### 4.1 BASELINES

We provide two major baseline comparisons to highlight the importance of VPD's compositional 3D architecture: SlotFormer and NeRF-dy. SlotFormer is a compositional 2D object-based video model (Wu et al., 2023) and NeRF-dy is a non-compositional 3D prediction model (Li et al., 2021a).

**SlotFormer** is a 2D video prediction model that operates over a slot-based latent representation (Wu et al., 2023). It uses a slot-based image encoder, SAVI (Kipf et al., 2021), and models how these slots

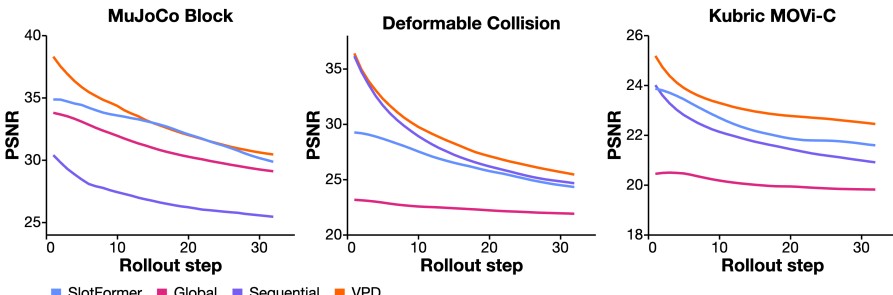

Figure 4: Video image quality as a function of rollout duration. VPD captures sharp detail, leading to high PSNR in the first steps of prediction. Low PSNR at the first steps indicates blurry reconstructions.

change over time with a transformer. In contrast to VPD, SlotFormer operates on single-view RGB camera data. To provide SlotFormer with the same data that VPD has access to, we treat each view in the multi-view dataset as a separate trajectory for training. Matching prior work, we condition SlotFormer on 6 history time-steps rather than 2 used for our model. We adapt the open-source code to run on the Kubric, MuJoCo and Deformable datasets.

**NeRF-dy** (Li et al., 2021a) is the most similar work to ours, with its ability to use multi-view video data and perform novel view synthesis. Since neither the code nor the datasets from that work are available, we perform ablations to VPD that roughly capture the key architectural differences between VPD and NeRF-dy as highlighted below:

1. **NeRF-dy uses a *global* latent vector per scene**. Instead of a 3D latent representation, NeRF-dy uses a single latent vector $z$ to represent the scene. This affects both the dynamics model (implemented as an MLP on $z$ instead of a GNN) and the renderer (globally conditioned on $z$). We implement this as an ablation, "Global", which uses the image and camera orientation encoder from Li et al. (2021a), and models the dynamics using an MLP rather than a GNN. As in VPD, the encoder and the dynamics are trained jointly.

2. **NeRF-dy uses *sequential* training of the encoder/decoder followed by dynamics**. NeRF-dy pretrains the encoder and decoder then holds their weights fixed and trains the dynamics model with supervision in the latent space. We use this training scheme in conjunction with the VPD particle-based architecture to investigate the significance of end-to-end training. This ablation is referred to as "Sequential".

More implementation details for these baselines are available in Appendix B and Appendix C.

## 5  RESULTS

Across the MuJoCo, Kubric, and Deformables datasets, we find that VPD produces detailed, physically consistent long-horizon rollouts (Section 5.1), supports compositional 3D editing and re-rendering of novel scenes (Section 5.2), and can learn simple dynamics with as few as 16 trajectories and 1-2 RGB-D cameras (Section 5.3). All metrics and visuals are from the held-out set. Videos of rollouts for each model, as well as generalization experiments, can be found on the project website.

### 5.1  VIDEO PREDICTION QUALITY

We first evaluate each learned simulator's ability to generate long horizon video predictions. After training, each model is rolled out to generate predicted videos for 32 timesteps into the future, which are compared to the ground truth future frames. VPD performs consistently well across all three datasets (Table 2) which is to be expected given the high quality qualitative rollouts (Figure 3). The Global representation struggles to capture multi-object scenes, which can be seen in its lower scores on the Kubric datasets and Deformable Collision. SlotFormer performs well on the solid-colored rigid bodies it was designed for (MOVi-A/B), but struggles to represent detailed textures (MOVi-C) or deformable objects when they come into contact with the floor. VPD is able to successfully represent both multi-object collision dynamics and object shape changes. Kubric's scenes pose a challenge for

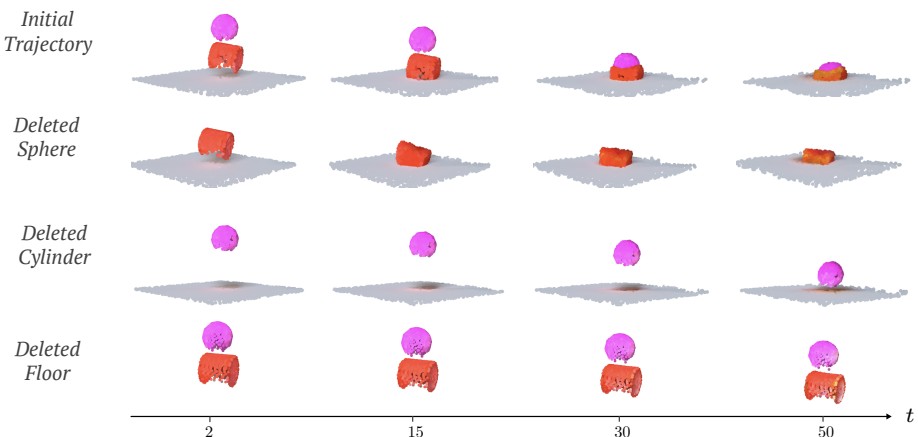

Figure 5: A demonstration of deleting various elements from our model's 3D scene representation on the Deformable Collision dataset before running the learned simulator forward for 50 time steps.

VPD because they have a huge diameter (100 meters); we find VPD predictions can be improved slightly by bounding the input scene (Appendix H).

VPD's prediction quality also degrades gracefully as a function of rollout step (Figure 4). VPD is able to represent significantly more detail than SlotFormer, as evinced by its high PSNR scores in the first few rollout steps. Meanwhile VPD's predictions at 32 steps are often better than the Global ablation at any step, and it remains much more accurate than the Sequential ablation at 32 steps.

These results highlight the ability of VPD's latent 3D particle representation to precisely capture a scene, and the locality bias of the VPD dynamics to preserve scene coherence over time.

## 5.2 EDITABILITY AND NOVEL VIEW GENERALIZATION

**3D point cloud editing**   When using a learned simulator with a 3D scene representation, it is possible to directly interrogate and even edit the scene before simulating. This capability could support many applications such as adding 3D assets to a 3D scene estimated from a set of sensors, tweaking object shapes, deleting objects, or moving them around.

We demonstrate 3D editing and simulation with VPD on a held-out test trajectory from the deformables collision dataset. Figure 5 visualizes the latent particles by applying the VPD renderer to each particle location to get a color, then plotting the colored points in 3D. In the original trajectory, a deformable pink ball and red cylinder fall onto a floor, squishing together into a pink and red lump. In order to edit the points corresponding to each object (ball, cylinder, floor) we roughly cluster the particles by position on the first frame. If we delete the pink ball from the inferred 3D scene, the cylinder still falls onto the ground. However, the top of the cylinder is no longer squished, since the pink ball did not come into contact with it. If we delete the cylinder, the pink ball falls all the way to the floor without deforming. If we delete the floor, the two objects both fall at a steady rate and neither one deforms since they both fall at the same acceleration. In all cases, this is roughly the behavior we would expect from the true underlying physics.

| Method | MuJoCo | Deformable | | | Kubric | | |
|---|---|---|---|---|---|---|---|
| | Block | Block | Multi | Collision | MOVi-A | MOVi-B | MOVi-C |
| SlotFormer | 32.541 | 28.673 | 27.545 | 23.145 | **30.934** | **28.810** | 22.367 |
| Global | 29.601 | 26.969 | 26.636 | 22.394 | 27.604 | 26.201 | 20.068 |
| Sequential | 25.858 | 30.854 | 31.256 | 27.911 | 28.613 | 25.921 | 21.854 |
| VPD (Ours) | **33.076** | **31.520** | **31.221** | **28.725** | 29.014 | 27.194 | **23.142** |

Table 2: PSNR scores averaged over 32 rollouts steps (higher is better).

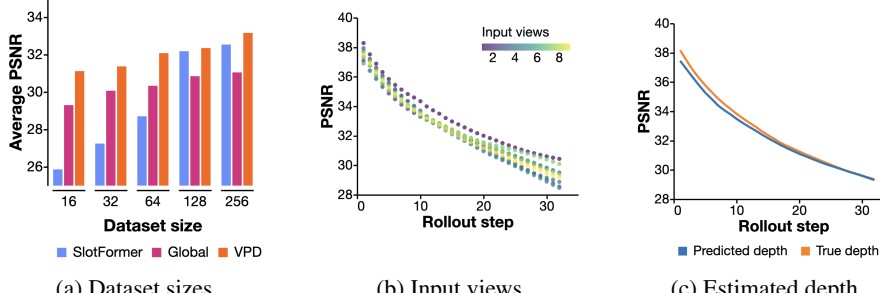

|                | (a) Dataset sizes | (b) Input views | (c) Estimated depth |

Figure 6: Video prediction results with various changes to the MuJoCo dataset that ablate different aspects of the input data. **(a)** VPD works effectively across a range of dataset sizes, whereas baselines require more data to achieve good results. **(b)** VPD can learn as well conditioned on one RGB-D camera as on 9 RGB-D cameras. **(c)** VPD is able to sustain reasonably high prediction quality even with an imprecise learned depth model (blue line).

**Novel viewpoint generalization**  Operating on 3D latent point clouds also allows us to render high-quality images from novel viewpoints, even when training on very few camera poses. In Figure 3, we show that VPD trained on the "Deformable Collision" dataset with only 4 fixed camera angles has no trouble generalizing to novel camera positions in a ring around the object. This sparse setting is challenging for e.g. NeRF-based methods which have weaker spatial biases, and impossible for 2D predictive models like SlotFormer. See Appendix Figure 9 for more examples.

## 5.3 DATA SPARSITY

In many applications, it might not be possible to obtain large numbers of RGB-D images of a particular scene at each timestep. We therefore investigate how well VPD performs when it has access to a very small number of trajectories or input views, and when it must use depth information predicted by a monocular depth estimation network.

**Different numbers of trajectories**  We generate datasets for MuJoCo Block of sizes 16, 32, 64, 128 and 256 trajectories, and evaluate each model's video prediction quality on the same set of 16 held-out trajectories (Figure 6a). Surprisingly, we find that even with as few as 16 distinct training trajectories, VPD achieves a PSNR of 31.22 which is only 1.85 points less than VPD trained with 256 distinct trajectories, and comparable in performance to baseline models trained with 4-8× as much data. This suggests that VPD could be suitable for applications in robotics where collecting many trajectories can be expensive.

**Different numbers of views**  We train models on MuJoCo Block with between 1 and 9 input views. With fewer views, VPD will have significant uncertainty about the object's shape (for example, it cannot see the back of the block). Despite this challenge, even with a single RGB-D camera VPD performs comparably to when it has 9 RGB-D cameras. While we are unsure of how VPD performs so well with a single RGB-D camera, one hypothesis is that since VPD's dynamics are trained end to end with the encoder, the dynamics could compensate for the particles that are missing due to self-occlusion. Future work will investigate whether these results apply to more complex datasets, and whether separately training the dynamics and the encoder erases this effect.

**Predicted depth**  While we use VPD with ground-truth depth information throughout our experiments, we provide a preliminary experiment using estimated depth instead. We extend the encoder's UNet with an additional output feature plane and interpret its value as pixel-wise metric depth. This depth prediction is supervised with an MSE loss on the training images and receives gradients via end to end training. On the MuJoCo Block dataset, VPD makes predictions of reasonable quality even with imprecise depth estimates (Figure 6c).

## 6 DISCUSSION

Visual Particle Dynamics (VPD) represents a first step towards simulators that can be learned from videos alone. By jointly learning a 3D latent point cloud representation, dynamics model for evolving that point cloud through time, and renderer for mapping back to image space, VPD does not require

state information in the form of object masks, object geometry, or object translations and rotations. Since VPD encodes particles in 3D, it also supports novel view generation and video editing through direct point cloud interactions. To our knowledge, this combination of 3D interpretability, editability, and simulation is unique among predictive models learned without physical supervision.

However, VPD also has limitations. As evidenced in our videos and figures, VPD struggles with blurring over very long rollouts, a common challenge in video prediction. It also requires access to RGB-D videos, which are not always available. While we showed a proof-of-concept for VPD working with predicted depth instead of ground-truth depth, this still requires a depth signal for training which may not always be available. Future work will need to investigate the potential application of pre-trained depth models with fine-tuning to a particular scenario.

We believe that VPD's ability to learn simulators directly from sensors has several important implications. For robotics, VPD could be used to learn a simulator directly from real data without requiring a separate state estimation step. This could then support much more effective sim-to-real transfer by learning a simulator that better reflects the robot's environment. For graphics, VPD could support new applications such as realistic video editing for systems where it is difficult to hand-code simulators. Overall, VPD opens new directions for how simulators can be learned and deployed when no analytic simulator is available.

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

# Appendix

## A   VPD IMPLEMENTATION DETAILS

This section describes details of our VPD implementation. Dataset-specific hyperparameters can be found in Table 3.

### A.1   ARCHITECTURES

**Encoder**   Figure 7 shows a schematic of the encoder network. The network uses a UNet architecture (Ronneberger et al., 2015) with the following numbers of channels: (64, 128, 256, 512, 512, 256, 128, 64, 16). The network outputs a 16-dimensional feature vector $\boldsymbol{Z}_{ij}$ for each input pixel, which we associate with the corresponding particle. When using the UNet for depth prediction, we add another output channel (17 total) and take this channel to be a pixel-wise depth prediction; the predicted depth is not included in the particle features $\boldsymbol{Z}$. There is a ReLU activation between each layer and no batch normalization. The UNet has a final softplus activation. The pixels from each image are unprojected together into the world coordinate frame, cropped to only include locations in the workspace (Table 3). They are then subsampled uniformly at random down to a set of $2^{14}$ particles representing a single time-step.

**Dynamics model**   The dynamics model is a hierarchical message passing graph neural network using MLPs for the node and edge encoders, updaters, and decoders. There is a separate set of parameters for each type of {node, edge} {encoder, updater, decoder}. We use 10 steps of message passing amongst the abstract nodes with un-tied weights at each message passing step. Each encoder or updater MLP has shape (128, 128), the location decoder MLP has shape (128, 128, 128, 3), and the feature decoder MLP has shape (128, 128, 128, 16), where 3 is the location delta and 16 is the particle feature delta. The updater MLPs use an output layer norm and an output residual connection.

The outputs of this GNN are one 3-d location delta vector and one 16-d feature delta vector for each current-timestep particle. The location delta goes into a $\tanh$ function and then is scaled by the "location scale" from Table 3, which helps early in training by preventing particles from moving unrealistically fast. The particles at the next timestep are produced by applying these deltas.

**Renderer**   The renderer is based on the jaxnerf codebase with accelerated Pallas CUDA kernels for nearest-neighbor finding. We use 16 approximate nearest neighbors for computing the features for input to the NeRF MLPs, and the kernel parameters are in Table 3. We use two tiers of sampling for NeRF rendering, a coarse layer with 128 sampling locations and no view directions and a fine layer with 128 (coarse) + 256 (fine) sampling locations. The coarse and fine networks have 8 layers of width 256 before concatenating view directions (if using), followed by 1 layer of width 128. Skip connections are added to every 4th layer and ReLU activations are used throughout. We use fourth-degree positional encodings on the view directions in the fine MLP.

### A.2   TRAINING

During training, we roll out the model for $T = 6$ time steps. For each unrolled time step, we render 256 rays, and supervise on the corresponding ground truth pixel value as in Equation 3. We apply a small amount of Gaussian noise centered at 0 and with sigma given by "location noise" in Table 3 to the particle locations during training rollouts to improve robustness. We use a batch size of 16 split across 16 GPUs. Optimization uses the Adam optimizer (Kingma & Ba, 2014) with a learning rate that begins at $3e - 4$, then decays by a factor of 3 at 100K and 300K updates. Models are trained for 400K updates.

### A.3   HIERARCHICAL MESSAGE PASSING

The hierarchical message algorithm is detailed in Algorithm 1. The initial features for the particle nodes are the features predicted by the UNet encoder, and the initial feature for the edges is the 3D

| Parameter | MuJoCo | Deformable | Kubric |
|---|---|---|---|
| $r_s$ | 0.1 | 1.0 | 1.0 |
| $r_s^a$ | 0.3 | 3.0 3.0 | |
| Input views | 1 | 4 | 9 |
| Kernel radii | (0.0, 0.05, 0.1, 0.5) | (0.0, 0.5, 1.0, 4.0) | (0.0, 0.1, 0.4, 2.0) |
| Kernel bandwidths | (0.05, 0.05, 0.05, 0.5) | (0.5, 0.5, 1.0, 4.0) | (0.1, 0.2, 0.4, 2.0) |
| Location scale | 0.1 | 0.5 | 0.5 |
| Location noise | $1e-5$ | $1e-3$ | $1e-4$ |
| Near plane | 0.3 | 9.0 | 0.5 |
| Far plane | 5.0 | 30 | 15 |
| Workspace min | [-1, -1, -1] | [-8, -8, -2] | [-10, -10, -10] |
| Workspace max | [1, 1, 1] | [8, 8, 8] | [10, 10, 10] |

Table 3: Hyperparameters for VPD by dataset

relative position between the start and end nodes. Abstract nodes start with no features, identified in Algorithm 1 as $\varnothing$.

---

**Algorithm 1:** Hierarchical message passing

---

**input :** Typed graph $\mathcal{G} = (\mathcal{V}, \mathcal{E})$
```
/* Encode particle nodes and edges.  */
```
$\mathcal{V}_{\mathcal{P}} \leftarrow \text{EncodeNode}(\mathcal{V}_{\mathcal{P}})$
$\mathcal{E}_{\mathcal{P} \rightarrow \mathcal{A}} \leftarrow \text{EncodeEdge}(\mathcal{E}_{\mathcal{P} \rightarrow \mathcal{A}})$

```
/* Update particle → abstract edges.  */
```
$\mathcal{E}_{\mathcal{P} \rightarrow \mathcal{A}} \leftarrow \text{UpdateEdge}(\mathcal{E}_{\mathcal{P} \rightarrow \mathcal{A}}, \mathcal{V}_{\mathcal{P}}, \varnothing)$

```
/* Update abstract nodes.  */
```
$\mathcal{V}_{\mathcal{A}} \leftarrow \text{UpdateNode}(\varnothing, \mathcal{E}_{\mathcal{P} \rightarrow \mathcal{A}})$

```
/* Perform message passing among abstract nodes.  */
```
**for** $k = 1, \ldots, K$ **do**
  $\quad \mathcal{E}_{\mathcal{A} \rightarrow \mathcal{A}} \leftarrow \text{UpdateEdge}(\mathcal{E}_{\mathcal{A} \rightarrow \mathcal{A}}, \mathcal{V}_{\mathcal{A}}, \mathcal{V}_{\mathcal{A}})$
  $\quad \mathcal{V}_{\mathcal{A}} \leftarrow \text{UpdateNode}(\mathcal{V}_{\mathcal{A}}, \mathcal{E}_{\mathcal{A} \rightarrow \mathcal{A}})$
**end**
```
/* Update features of abstract → current particle nodes.  */
```
$\mathcal{E}_{\mathcal{A} \rightarrow \mathcal{P}^t} \leftarrow \text{UpdateEdge}(\mathcal{E}_{\mathcal{A} \rightarrow \mathcal{P}^t}, \mathcal{V}_{\mathcal{A}}, \mathcal{V}_{\mathcal{P}^t})$

```
/* Update current-time particle nodes.  */
```
$\mathcal{V}_{\mathcal{P}^t} \leftarrow \text{UpdateNode}(\mathcal{V}_{\mathcal{P}^t}, \mathcal{E}_{\mathcal{A} \rightarrow \mathcal{P}^t})$

```
/* Decode predicted locations and features.  */
```
$\Delta \boldsymbol{X} \leftarrow \text{DecodeLocation}(\mathcal{V}_{\mathcal{P}^t})$
$\Delta \boldsymbol{Z} \leftarrow \text{DecodeFeature}(\mathcal{V}_{\mathcal{P}^t})$

---

## B   SLOTFORMER TRAINING

We experimented with both the OBJ3D and CLEVRER hyperparameter configurations for all datasets, and found the results from the OBJ3D configuration to be superior. These hyperparameters are in Table 4.

Kubric datasets use 8 slots while the Mujoco and Deformable datasets use 4 slots (8 slots caused instabilities in training). Kubric was also downsampled to $64 \times 64$ resolution for training due to training stability challenges. MuJoCo and Deformables datasets were trained at $128 \times 128$ resolution.

After training, the slots learned by SAVi are extracted for the training and validation datasets. SlotFormer is then trained on these slot representations, using the hyperparameters in Table 5.

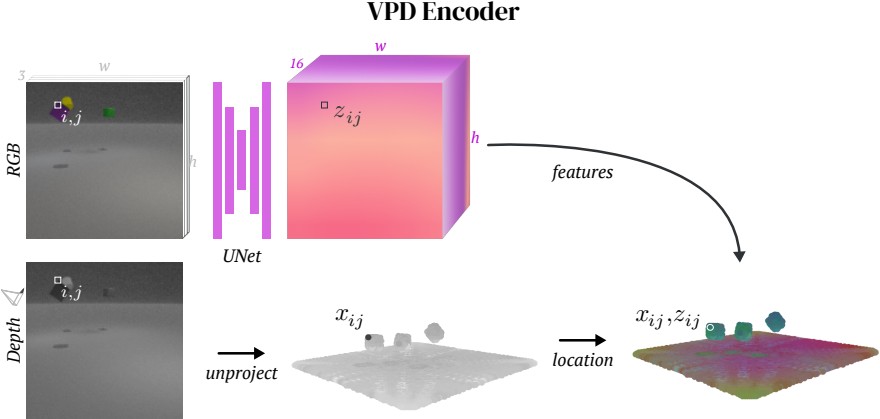

Figure 7: A diagram of the VPD Encoder. The encoder maps an RGB-D image into a latent point cloud representation. Each pixel observed from a given camera is unprojected into 3D using the depth information. The RGB image is fed into a UNet to produce per-pixel latent features.

| Hyperparameter | Setting |
|---|---|
| epochs | 40 |
| learning rate | $1e-4$ |
| gradient clip | 0.05 |
| history length | 6 |
| frame offset | 1 |
| batch size | 64 |
| slot mlp size | 256 |
| num iterations | 2 |
| encoder channels | (3, 64, 64, 64, 64) |
| encoder ks | 5 |
| encoder out channels | 128 |
| decoder channels | (128, 64, 64, 64, 64) |
| decoder resolution | (8, 8) |
| decoder ks | 5 |
| predictor type | transformer |
| pred_norm_first | True |
| pred_rnn | True |
| pred_num_layers | 2 |
| pred_num_heads | 4 |
| pred_ffn_dim | $128 \times 4$ |
| use_post_recon_loss | True |
| kld_method | none |
| post_recon_loss_w | 1 |
| kld_loss_w | $1e-4$ |

Table 4: SAVi training hyperparameters

## C    ABLATION IMPLEMENTATION DETAILS

### C.1    SEQUENTIAL

Supervising the dynamics in the particle-based latent space is more complex than it is with the NeRF-dy architecture, since particles have no correspondences across timesteps. To avoid the correspondence issue, we randomly sample $2^{14}$ 3D locations in the scene's workspace, then compute the renderer input features at those points under (1) the particle cloud at the next timestep predicted by the dynamics model and (2) the particle cloud from the encoder obtained from the set of RGB-

| Hyperparameter | Setting |
|---|---|
| epochs | Kubric: 15, Deformables: 80, MuJoCo: 80 |
| learning rate | $2e-4$ |
| history length | 6 |
| sample frames | 16 (10 rollout) |
| frame offset | 1 |
| batch size | 64 |
| slot size | 128 |
| t_pe | $sin$ |
| d_model | 128 |
| num layers | 4 |
| num heads | 8 |
| ffn_dim | $128 \times 4$ |
| norm_first | True |
| decoder channels | (128, 64, 64, 64, 64) |
| decoder resolution | (8, 8) |
| decoder ks | 5 |
| use_img_recon_loss | True |
| rollout length | 10 |
| slot_recon_loss_w | 1 |
| img_recon_loss_w | 1 |

Table 5: SlotFormer training hyperparameters

D images at the next timestep. The loss is then the mean squared error between the features at corresponding locations. This ensures that the dynamics model's predictions lead to renderer input features which are correct at all locations in the scene. In the limit of perfect predictions, this would translate to correct reconstruction.

We find sequential training to be less robust than end-to-end training and needed to use lower learning rates: $1e-4$ to start for both pre-training and dynamics training, decaying by a factor of 10 followed by an additional factor of 3. We perform 100K steps of auto-encoder pre-training and 400K steps of dynamics model training.

## D  SCALING WITH NUMBER OF POINTS

We find that VPD's video quality increases as more particles are used to represent the scene. A greater number of particles enables the model to represent details of geometry and textures.

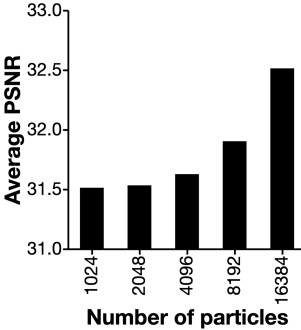

Figure 8: Evaluating the performance of VPD with varying numbers of particles, measured as average PSNR over a 32-step rollout. More particles result in consistently better performance.

# E    SSIM RESULTS

SSIM is an alternative metric to PSNR for evaluating image similarity. It compares statistics of windowed pixels and permits more blurring than the pixel-wise error of PSNR. Results are broadly similar to scoring with PSNR, but with less discriminitive power due to SSIM's insensitivity to detail.

| Method | MuJoCo | Deformable | | | Kubric | | |
|---|---|---|---|---|---|---|---|
| | Block | Block | Multi | Collision | MOVi-A | MOVi-B | MOVi-C |
| SlotFormer | **0.954** | **0.957** | **0.956** | 0.913 | **0.880** | **0.867** | 0.547 |
| Global | 0.954 | 0.936 | 0.932 | 0.885 | 0.848 | 0.846 | 0.529 |
| Sequential | 0.699 | 0.953 | 0.953 | 0.933 | 0.850 | 0.845 | 0.606 |
| VPD (Ours) | **0.960** | **0.957** | 0.953 | **0.939** | 0.857 | **0.859** | **0.703** |

Table 6: SSIM scores averaged over 32 rollouts steps (higher is better). Ties awarded within 0.01.

# F    RENDERING FROM NOVEL VIEWS

Since VPD contains a 3D representation of the scene and learns a ray-based renderer, we can render novel views from camera positions not seen in training. Figure 9 shows an example of this for the Deformable Collisions dataset on held-out trajectories.

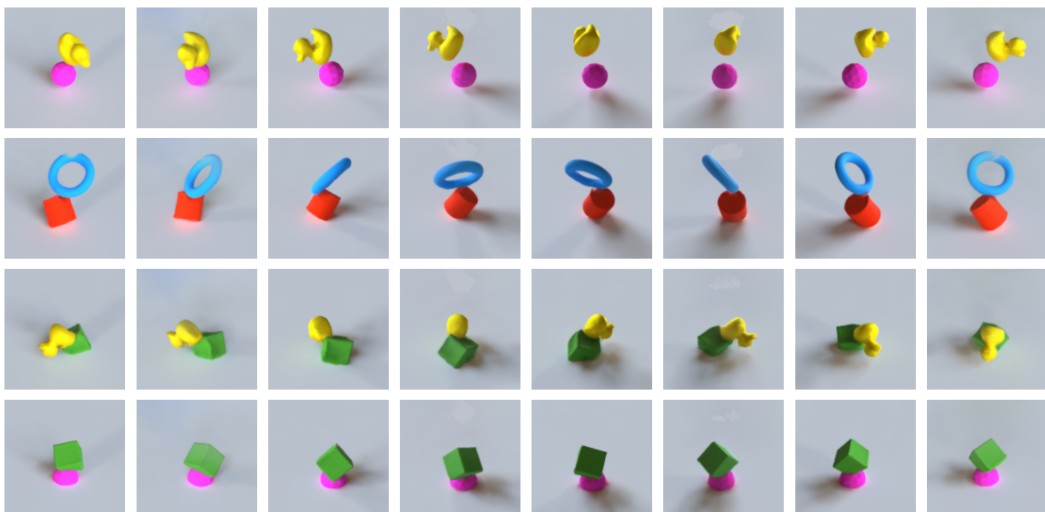

Figure 9: Generalization over camera pose. A VPD model trained on Deformable Collision is rolled out on the test set for 15 frames, and the final latent state is rendered from multiple camera positions oriented in a ring around the objects. A video example of free camera movement can be found [here].

# G    ROLLOUT COMPARISONS BETWEEN VPD AND BASELINES

In each of these diagrams, the first two frames are ground-truth inputs to the models. SlotFormer is conditioned on six frames, not two; the first four frames of SlotFormer conditioning are not shown. All videos are from the held-out evaluation set.

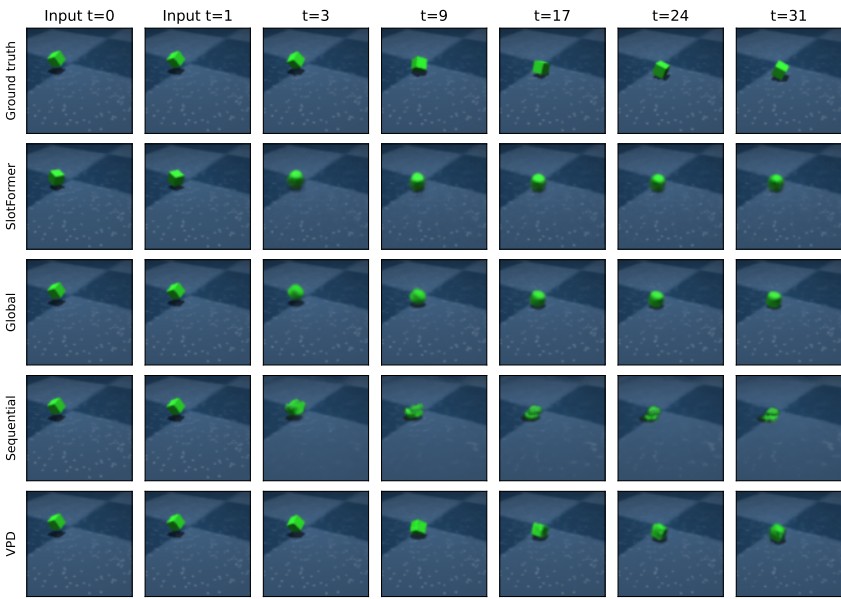

Figure 10: Rollout comparison on MuJoCo Block dataset

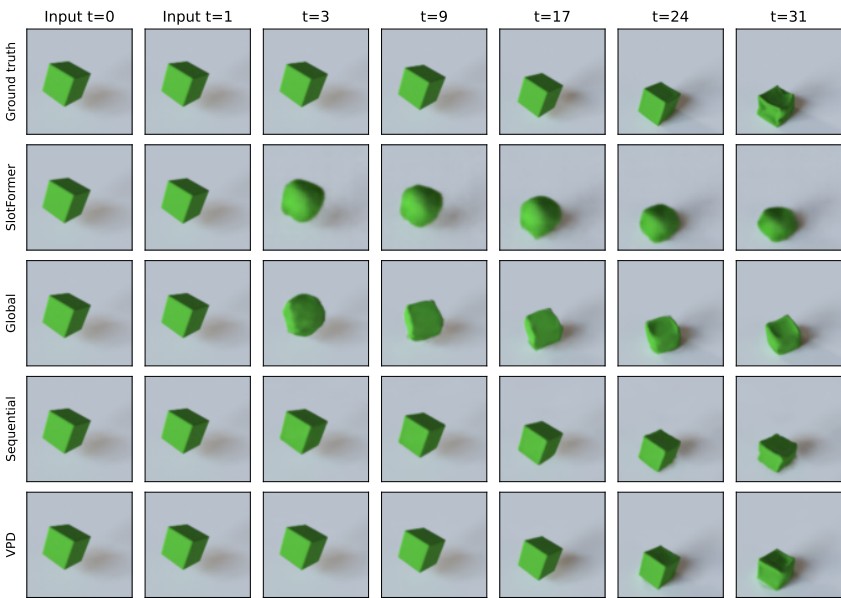

Figure 11: Rollout comparison on Deformable Block dataset

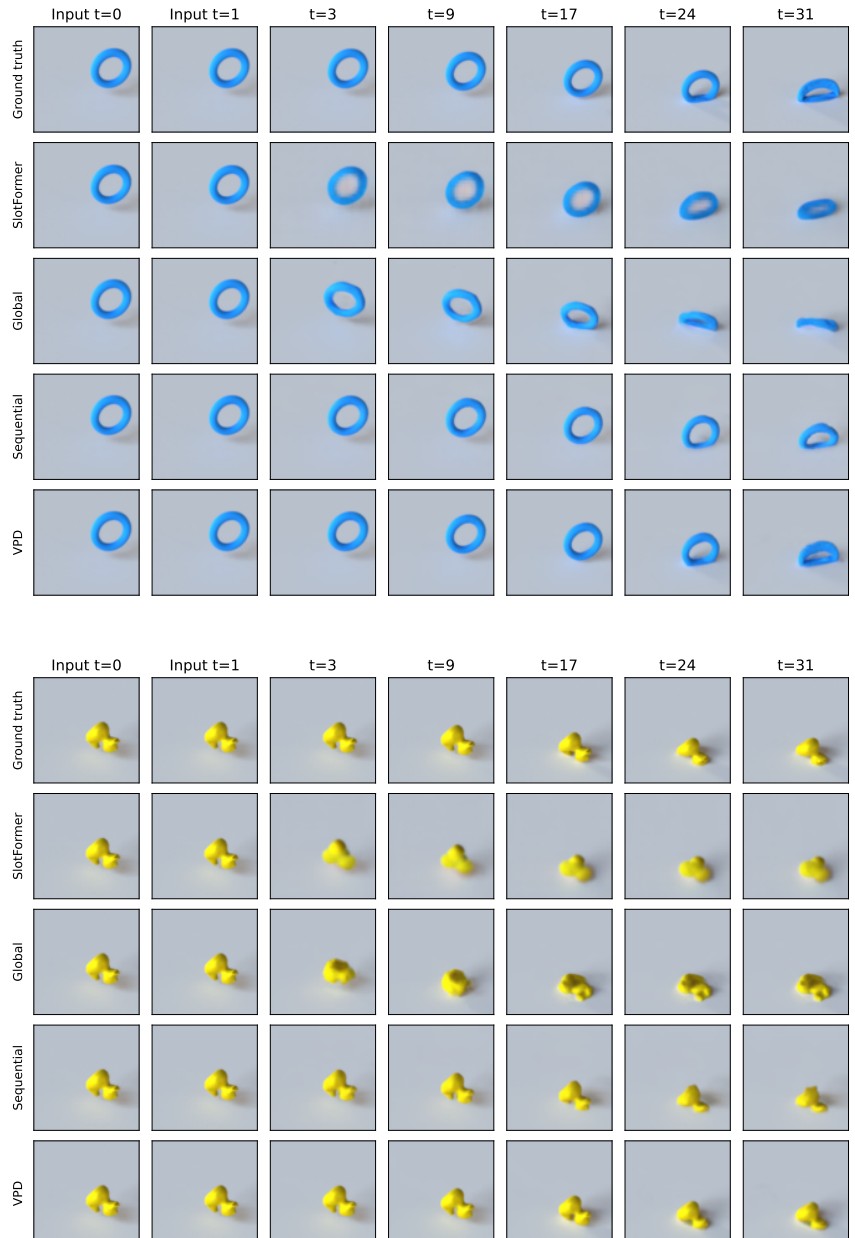

Figure 12: Rollout comparison on the Deformable Multi dataset.

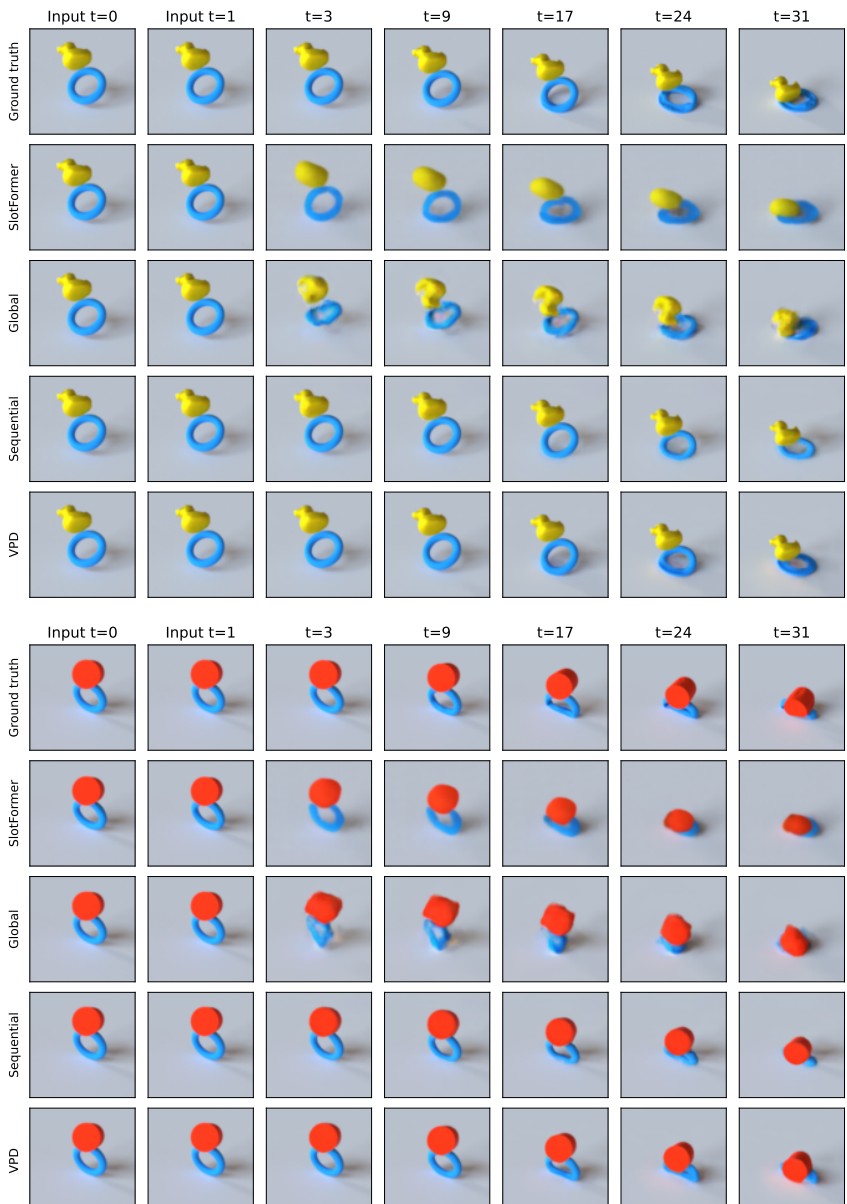

Figure 13: Rollout comparison on Deformable Collision dataset.

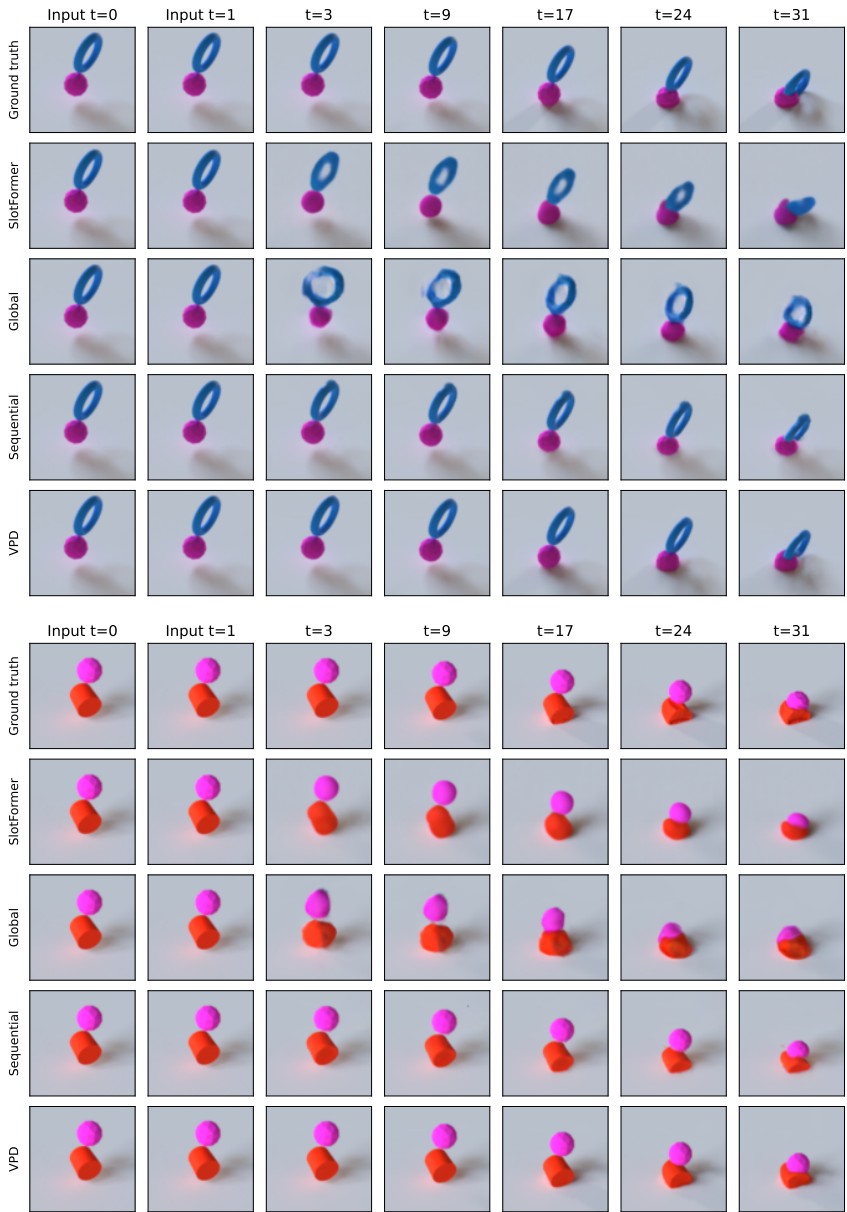

Figure 14: Rollout comparison on Deformable Collision dataset.

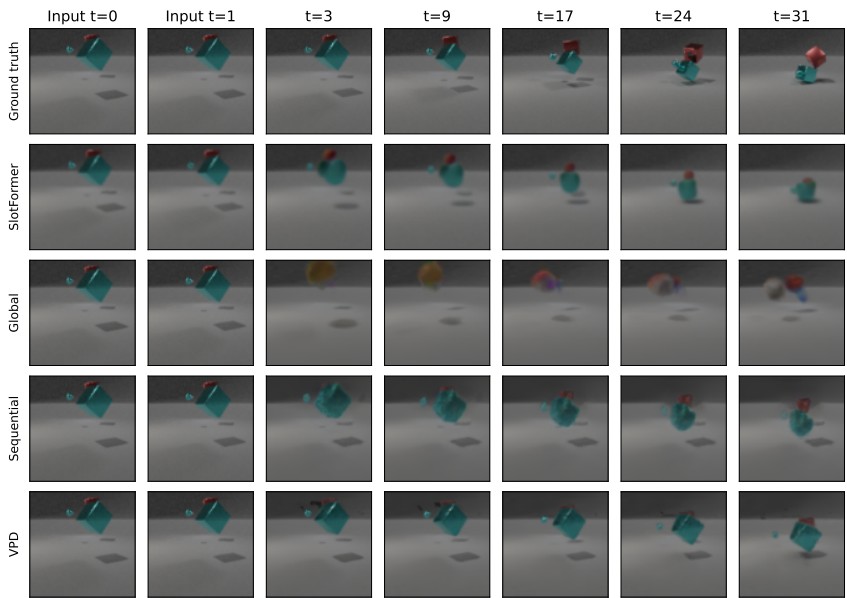

Figure 15: Rollout comparison on Kubric MOVi-A dataset.

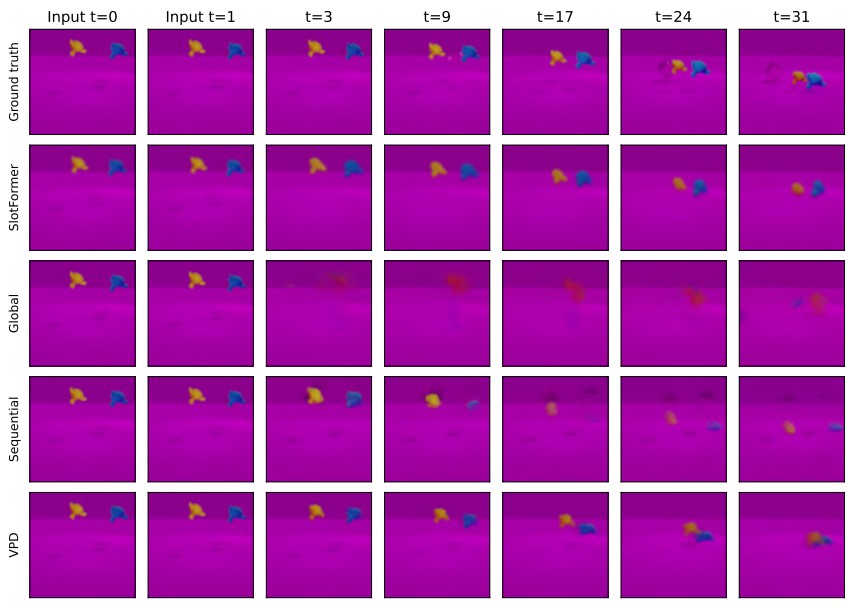

Figure 16: Rollout comparison on Kubric MOVi-B dataset.

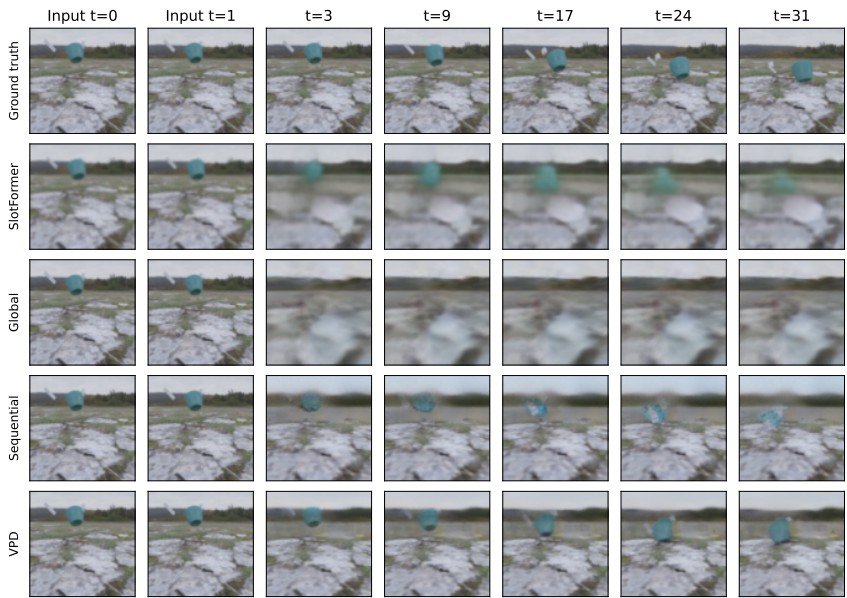

Figure 17: Rollout comparison on Kubric MOVi-C dataset.

## H  IMPROVING VPD RECONSTRUCTION QUALITY ON KUBRIC.

In the main text we use the Kubric dataset with the conventional camera setup, where the cameras point sideways and cover the walls surrounding the scene. However, scenes with far-away objects are generally challenging for NeRFs, as they require sampling points at lower resolution along the ray, resulting in blurry reconstructions.

We generate a version of Kubric where the cameras are positioned higher and point downwards, creating a bounded scene that is more suitable for NeRFs. Figure 18 demonstrates that the quality of VPD reconstructions considerably improves on the updated camera setup. The objects are more crisp and track the ground-truth object trajectory more accurately.

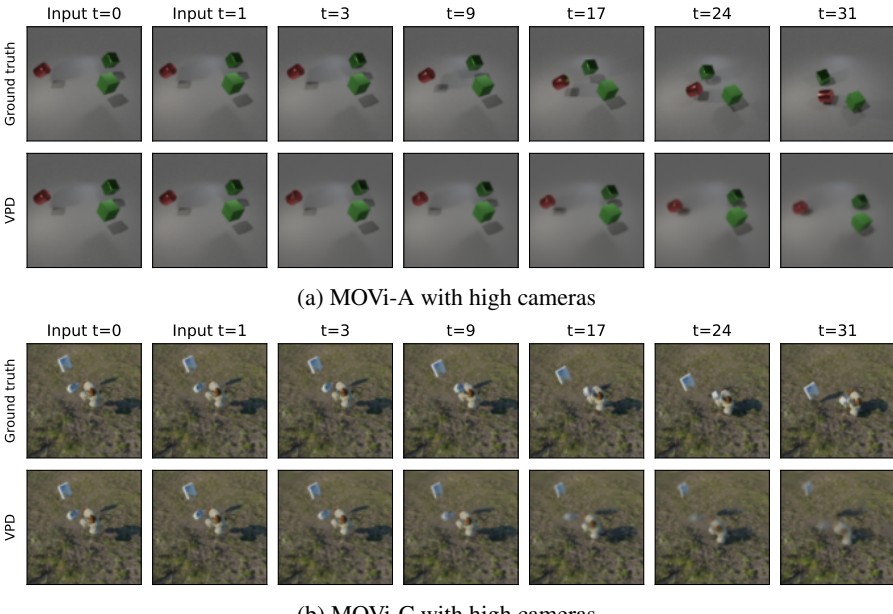

(a) MOVi-A with high cameras

(b) MOVi-C with high cameras

Figure 18: VPD recontructions on Kubric with the modified camera setup.

# I  FEATURE ENCODING EXPERIMENTS

The particle features from VPD resemble those used in PointNerf (Xu et al., 2022), although there are some differences. PointNerf includes an extra feature MLP $F(f_k^i, \boldsymbol{x} - p_x^i)$. This MLP processes nearby points (within a certain range $k$) to predict the feature pattern at a specific $\boldsymbol{x}$. We experimented with a modified version of VPD that uses a similar additional feature encoder $F$, but had to reduce the number of prediction steps to $T = 2$ to fit in memory. We did not observe any significant benefits when compared to the full VPD model, particularly when considering longer rollouts (Figure 19a).

Another variation of feature encoding is the normalization feature described in PAPR (Zhang* et al., 2023). We explored a variation of VPD that normalizes the feature (Equation (1)) contributions with respect to the relative distances between the query point $\boldsymbol{x}$ and a given nearby particle $p_x^i$. Similarly, we did not observe any notable advantages compared to the complete VPD model (Figure 19b).

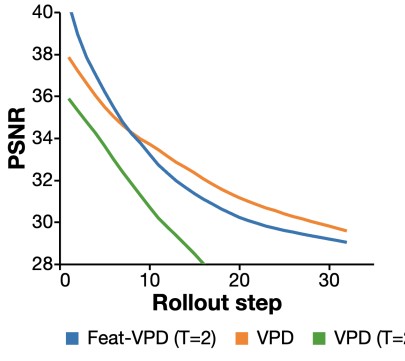
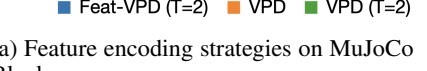
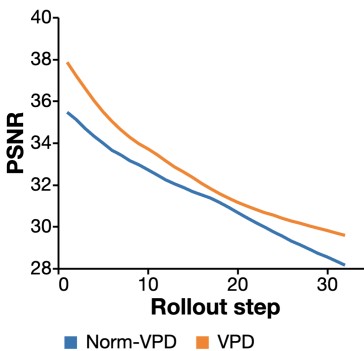

(a) Feature encoding strategies on MuJoCo Block.

(b) Feature normalization strategies on MuJoCo Block.