# OpenReview forum: "Learning 3D Particle-based Simulators from RGB-D Videos"
_ICLR.cc/2024/Conference — ICLR 2024 poster_

### Official Review · Reviewer_8vFC · 2023-10-21

**Soundness:** 3 good
**Presentation:** 3 good
**Contribution:** 3 good
**Rating:** 8
**Confidence:** 3

**Summary:**

The proposed paper introduces a novel approach for learning simulations from videos. The approach relies on RGB-D images as input and uses a combination of neural networks (UNet, GraphNetwork, Renderer) to first generate a latent particle representation that is then rendered into output images.

**Strengths:**

- The paper addresses an important research direction -- learning physics directly from videos is a challenging and open problem.
- To my knowledge, the proposed model and training procedure (with RGB-D images) is novel.
- The proposed method seems technically sound and the results are convincing.
- The paper is mostly well-written and easy to follow.

**Weaknesses:**

- Some sections of the text could provide more details (more details below).
- Only synthetic data was used to test the method.
- Not all experiments are convincing (more details below).

**Questions:**

- Section 3.1.: It is not clear how the per-pixel latent features are generated. Ronneberger et al. uses down convolutions to produce a compressed latent space, which would not result in a per-pixel latent. How does the UNet architecture used in this paper differ from the original one?

- Section 3.1. This section is difficult to follow as some term are not defined or introduced properly. E.g. what is a 'conditioning set'? What is a "work space" and how is it defined?

- While the paper mentions 'blurring over very long rollouts', it does not provide a discussion on what this actually means. After how many time steps do outputs become blurry? Can this be quantified? What about baseline methods?

- The authors may also want to discuss the following paper: H. Shao, T. Kugelstadt, T. Hädrich, W. Pałubicki, J. Bender, S. Pirk, D. L. Michels, Accurately Solving Rod Dynamics with Graph Learning, Conference on Neural Information Processing Systems (NeurIPS), 2021

**Details Of Ethics Concerns:**

No ethical concerns regarding this submission.

---

> ### Author Response · Authors · 2023-11-22
> **Reply**
>
> We thank the reviewer for their time and detailed comments. We are glad that the reviewer believes our paper is solving an important open problem with a novel, sound technique. We also appreciate the feedback that will help make the paper clearer.
>
> > Section 3.1.: It is not clear how the per-pixel latent features are generated. Ronneberger et al. uses down convolutions to produce a compressed latent space, which would not result in a per-pixel latent. How does the UNet architecture used in this paper differ from the original one?
>
> The original UNet architecture (Ronneberger et al.) first uses convolutions and pooling to create a smaller feature image with more channels, then uses skip connections and up-convolutions to create per-pixel labels (their Fig. 1). It has subsequently been used for a variety of other image-to-image tasks, e.g. conditional generative models [1]. Our UNet implementation differs mainly on the specific number of channels we use [64, 128, 256, 512, 512, 256, 128,64, 16]  (Appendix A.1). We have added a small modification in Figure 7 to include the input/output dimensionality.
>
> > Section 3.1. This section is difficult to follow as some term are not defined or introduced properly. E.g. what is a 'conditioning set'? What is a "work space" and how is it defined?
>
> We thank the reviewer for pointing this out. We have edited the document to remove/define these terms.
>
> > While the paper mentions 'blurring over very long rollouts', it does not provide a discussion on what this actually means. After how many time steps do outputs become blurry? Can this be quantified? What about baseline methods?
>
> We apologize for the confusion. The PSNR metric captures blurring, and is plotted as a function of rollout step in Figure 4. From this figure, you can see that PSNR falls over the course of the rollout. However, the PSNR decreases more slowly for VPD relative to the baselines. This is further captured in the videos on the website: https://sites.google.com/view/latent-dynamics.
>
> > The authors may also want to discuss the following paper: H. Shao, T. Kugelstadt, T. Hädrich, W. Pałubicki, J. Bender, S. Pirk, D. L. Michels, Accurately Solving Rod Dynamics with Graph Learning, Conference on Neural Information Processing Systems (NeurIPS), 2021
>
> Thank you for bringing our attention to this paper. We have added it to our introduction which you can see in the revised paper.
>
> [1] Isola, P., Zhu, J., Zhou, T., & Efros, A.A. (2016). Image-to-Image Translation with Conditional Adversarial Networks. 2017 IEEE Conference on Computer Vision and Pattern Recognition (CVPR), 5967-5976.

---

### Official Review · Reviewer_HnBD · 2023-10-24

**Soundness:** 3 good
**Presentation:** 3 good
**Contribution:** 3 good
**Rating:** 6
**Confidence:** 4

**Summary:**

In this paper, the authors proposed a neural-network-based 3D simulator where a graph neural network is used to estimate the 3D dynamics. The simulator transforms the multi-view RGBD images into a latent point cloud and uses a differentiable simulator to transform the simulated result into images, such that an image-based loss is used to train the simulator end-to-end. The proposed simulator supports 3D state editing, novel view re-rendering and multi-material simulation.

**Strengths:**

The paper uses a explicit point cloud representation, such that it supports scene editing, novel-view re-rendering. The experimental results have shown the effectienss of the method in such applications.

**Weaknesses:**

The claim that existing learned world models are not simulators is not accurate. The function of simulator is to estimate the transition model of the system.
The main weakness of the proposed simulator is that it does not support the simulation of the environment response to different actions. Therefore, it can only predict the future state of a passive system.
Secondly, the author does not validate the extrapolation ability, which is important for a simulator. Experiments on out-of-distribution initial state can be added.

**Questions:**

Can the proposed simulator support different physics properties, such as density, friction?

---

> ### Author Response · Authors · 2023-11-22
> **Reply**
>
> We thank the reviewer for taking the time to read the paper and providing constructive comments.
>
> > The claim that existing learned world models are not simulators is not accurate. The function of simulator is to estimate the transition model of the system.
>
> We apologize for any confusion about the use of the word simulator. When we use the word simulator, we are referring to a model that can be used similarly to an analytic simulator such as PyBullet, MuJoCo, or those in graphics. These simulators need to do more than estimate the transition model of the system. They need to support composing new scenarios, rendering from new viewpoints, and modeling larger scenes than seen in training. It is these qualities that most prior work in this area lack, or require additional supervision in order to achieve.
>
> > Secondly, the author does not validate the extrapolation ability, which is important for a simulator. Experiments on out-of-distribution initial state can be added.
>
> In a further experiment, we evaluate the model’s extrapolation ability by running it on scenes with up to 32 deformable objects instead of the 2 seen in training. The model produces realistic dynamics for this much more complex scene, as can be seen in new videos on the website: https://sites.google.com/view/latent-dynamics#h.sckus2cmnoec. We will add this experiment to the camera-ready version of the paper if accepted.
>
> > Can the proposed simulator support different physics properties, such as density, friction?
>
> The simulator should be able to model objects with varying physical properties as long as those properties can be discriminated from visual observations. To demonstrate this point, we performed a new experiment on a dataset with objects with two different restitutions (how squishy the object is). VPD can infer this difference in elasticity based on the colors of the objects, and makes realistic predictions for how the objects should move. We have added this experiment to the website: https://sites.google.com/view/latent-dynamics#h.13u38bw637iq. We will include it in the camera-ready version of the paper if accepted.
>
> > The main weakness of the proposed simulator is that it does not support the simulation of the environment response to different actions. Therefore, it can only predict the future state of a passive system.
>
> While it is true that the current version of VPD does not support actions, this is not, in principle, difficult to do. Prior work  has shown how actions can be used with graph neural networks to support robotics [1], and similar approaches should be possible here. We believe this is a very interesting direction for future work.
>
> [1] Sanchez-Gonzalez, Alvaro, et al. "Graph networks as learnable physics engines for inference and control." International Conference on Machine Learning. PMLR, 2018.

---

### Official Review · Reviewer_CLWA · 2023-10-29

**Soundness:** 3 good
**Presentation:** 2 fair
**Contribution:** 3 good
**Rating:** 6
**Confidence:** 3

**Summary:**

The paper introduces VPD, a method to learn particle simulators given multiview RGBD videos. The setup is similar to NeRF-dy (Li et al.). The major difference is an explicit point-based neural simulation and rendering pipeline that avoids learning mappings between images and latent states.

VPD is evaluated on dastsets with deformable and rigid objects, showing better performance than SlotFormer (a video prediction method) and re-implemented NeRF-dy (a ) that roughly matches the paper.

**Strengths:**

**Presentation**
- The paper is generally easy to follow, and the video results on the website are helpful.

**Method**
- The point-based representations enable 3D editing capability during/before simulation.
- The use of PointNeRF is a nice design choice that couples particle-based simulation and differentiable rendering.

**Weaknesses:**

**Writing**
- The idea of learning dynamics/physics simulators from videos is not particularly new (e.g., NeRF-dy, [A-B]), but the intro and related work are positioned in a way that appears those works are not relevant. To give the readers sufficient context, I would recommend putting more effort into discussing the existing "video->simulator" works, their limitations, and the key differences in this work.


**Efficiency**
- One advantage of using a latent representation for simulation learning (e.g., NeRF-dy) is its efficiency, especially when the number of particles is large. Given enough training data, one would expect a latent representation to be more performant.

**Motivation**
- To obtain better physics simulators (of system dynamics), what do we gain by learning from videos? The motivation and evidence could be made more concrete.
  - Specifically, the problems mentioned in the intro seem can be solved by system identification with a differentiable simulator, e.g., Taichi, Warp, Brax, dojo, which can find physical parameters (e.g., friction coefficient) from input videos without re-learn how to simulate. One example is Le Cleac’h et al., 2023.
  - If the goal is to speed up the simulation, it seems distilling physics-based simulators into a neural architecture is a strong competitor.
- Overall, it would be great if the paper could give a compelling example where VPD outperforms (or has the hope to outperform) the physics-based simulator. One aspect might be the generality. Another case is when there are complex disturbances in the environment that cannot be modeled by the physics simulator.

**Rigid objects**
- It seems the results on rigid objects (Kubric Movi-a) contain undesirable deformations. This seems to suggest that the learned GNN is not able to enforce rigidity constraints for high-stiffness materials.
- It would be nice to show how the method works on more challenging cases, such as the collision between rigid objects (or a rigid object and the ground). Those cases typically require a small timestep for physics-based simulation.


[A] Qiao, Yi-Ling, Alexander Gao, and Ming Lin. "Neuphysics: Editable neural geometry and physics from monocular videos." Advances in Neural Information Processing Systems 35 (2022): 12841-12854.

[B] Heiden, Eric, et al. "Inferring articulated rigid body dynamics from rgbd video." 2022 IEEE/RSJ International Conference on Intelligent Robots and Systems (IROS). IEEE, 2022.

**Questions:**

N.A.

---

> ### Author Response · Authors · 2023-11-22
> **Reply 1 / 2**
>
> We thank the reviewer for their time and their helpful comments.
>
> > The idea of learning dynamics/physics simulators from videos is not particularly new (e.g., NeRF-dy, [A-B]), but the intro and related work are positioned in a way that appears those works are not relevant.
>
> Thank you for the feedback. We certainly did not intend to make it seem as if these works are not relevant, and of course NeRF-dy in particular informed our main ablations. We have updated the introduction and related work to add more detailed discussion of these works and the differences between them and VPD.
>
> > To obtain better physics simulators (of system dynamics), what do we gain by learning from videos?
>
> We gain two main things: (1) the possibility of customizing simulators to particular physics, (2) learning physics for systems that we do not understand sufficiently well to make analytic simulators for. For point (1): in theory, customization can be supported by system identification. However, previous work has suggested that it can be difficult to model certain real systems (such as those with friction or contact dynamics) with system identification (e.g. [1, 2])) especially when perception from a video is necessarily noisy and imperfect. Using a learned simulation technique instead can overcome these perceptual errors. For point (2): there are some physical dynamics that are extremely difficult to model analytically. Friction is a famous example – there are almost imperceptible ways in which the surfaces of objects end up affecting the way that their surface interact. Writing an analytic simulator is therefore not possible in these cases, so we may want to learn one from watching a video.
>
> > Specifically, the problems mentioned in the intro seem can be solved by system identification with a differentiable simulator, e.g., Taichi, Warp, Brax, dojo, which can find physical parameters (e.g., friction coefficient) from input videos without re-learn how to simulate.
>
> Indeed, SysID is a common technique to estimate parameters in combination with DiffSims. However, system identification from video poses additional challenges in obtaining 3D models of objects and tracking them accurately. For example, Le Cleac'h [3] and NeuPhys [A] and [B], perform sysID on a scene with a single object. Moreover, these often involve either a relatively reduced set of parameters to be estimated [A], or can require binary object masks, knowledge of geometry [B] or 3D models learned ahead of time from a collection of still images [3],  none of which are necessary for VPD. In contrast, we were motivated by prior work on learned simulation which found that graph network simulators can outperform analytic simulators with sysID on ground-truth states [1, 2]. This can be especially promising for rigid contacts, which as you observe require small timesteps for analytic simulators.
>
> We have added further discussion of the papers referenced in this review to our related work.
>
> > If the goal is to speed up the simulation, it seems distilling physics-based simulators into a neural architecture is a strong competitor.
>
> In effect, this is what Mesh Graph Networks [4] do (and other learned/hybrid simulators). They distill physics-based simulators into graph neural networks, which results in computational speed-ups of 100-1000x. In this paper, we show how to use similar graph neural network architectures for physical dynamics, but adapted to work from RGB-D videos. In principle, this then allows us to distill physics simulators for any system that we can capture RGB-D video from.
>
> > Overall, it would be great if the paper could give a compelling example where VPD outperforms (or has the hope to outperform) the physics-based simulator. One aspect might be the generality. Another case is when there are complex disturbances in the environment that cannot be modeled by the physics simulator.
>
> Prior work on learned simulation has found that graph network simulators can outperform physics-based simulators with sysID on ground-truth states for rigid body dynamics (specifically, cube-ground collisions, and planar pushing  [1, 2]). We demonstrate VPD on the same cube-ground collisions here, with the additional complexity that VPD learns from RGB-D videos rather than from state information. Given that the underlying dynamics architecture is similar to [1, 2], we expect that VPD would similarly outperform system identification with a physics-based simulator for these cases. Furthermore, relative to Brax and dojo, VPD is more general – it can operate over non-rigid dynamics, which we show in the deformable experiments.

---

> ### Author Response · Authors · 2023-11-22
> **Reply 2 / 2**
>
> > ​​It seems the results on rigid objects (Kubric Movi-a) contain undesirable deformations. This seems to suggest that the learned GNN is not able to enforce rigidity constraints for high-stiffness materials.
>
> The Kubric datasets are challenging for a variety of reasons, including their visual complexity and the very large scenes (~100 meters in diameter), which are challenging to cover with particles. The MuJoCo Block dataset consists of a rigid block hitting the ground at speed, and our results there demonstrate that the GNN is able to maintain rigidity. We have added more videos of this case to the site: https://sites.google.com/view/latent-dynamics#h.9d3hhg3usjkt. Prior work focused on this question found similar results [1].
>
> > It would be nice to show how the method works on more challenging cases, such as the collision between rigid objects (or a rigid object and the ground).
>
> The MuJoCo and Kubric datasets all consist of collisions among rigids and between rigids and the ground. We have added an additional experiment with single objects of different shapes interacting with the ground, some of them deformable and some of them fully rigid (https://sites.google.com/corp/view/latent-dynamics#h.13u38bw637iq). The results suggest that even with different object shapes, object-ground collisions are well represented with the VPD model.
>
>
> [1]: Allen, K.R., Lopez-Guevara, T., Rubanova, Y., Stachenfeld, K.L., Sanchez-Gonzalez, A., Battaglia, P.W., & Pfaff, T. (2022). Graph network simulators can learn discontinuous, rigid contact dynamics. Conference on Robot Learning.
>
> [2]: Allen, K.R., Rubanova, Y., Lopez-Guevara, T., Whitney, W.F., Sanchez-Gonzalez, A., Battaglia, P.W., & Pfaff, T. (2023). Learning rigid dynamics with face interaction graph networks. International Conference on Learning Representations.
>
> [3]: Cleac'h, S.L., Yu, H., Guo, M., Howell, T.A., Gao, R., Wu, J., Manchester, Z., & Schwager, M. (2022). Differentiable Physics Simulation of Dynamics-Augmented Neural Objects. IEEE Robotics and Automation Letters, 8, 2780-2787.
>
> [4] Pfaff, T., Fortunato, M., Sanchez-Gonzalez, A., & Battaglia, P. W. (2020). Learning mesh-based simulation with graph networks. International Conference on Learning Representations.

---

### Official Review · Reviewer_cZVt · 2023-10-29

**Soundness:** 3 good
**Presentation:** 3 good
**Contribution:** 2 fair
**Rating:** 6
**Confidence:** 3

**Summary:**

This paper proposes a method to learn a particle-based representation, a neural simulator, and a neural renderer from posed RGB-D videos.

First, this method projects pixels back to 3D point clouds with known depth values and camera poses. Second, a UNet is applied to the RGB channels to extract per-pixel features. Then, a neural simulator implemented using MeshGraphNet is applied to infer the dynamics of the scene. In the end, a NeRF-style neural renderer is used to synthesize the image, where the feature of a query point is computed by filtering over neighboring points.

With ground truth RGB-D video sequences, this pipeline can learn the feature extractor, neural simulator, and neural renderer end-to-end. Experiments show that it can handle different dynamics (rigid, soft, contact, etc.). Ablation studies suggest that the designed components are necessary for the entire system.

**Strengths:**

The motivation and building blocks of this paper are clear and reasonable. The choices of neural simulator and renderer are up-to-date. With a general, unstructured, and differentiable simulator (GNN) and render (NeRF), it is easier to simultaneously train this system end-to-end.

Moreover, training the system end-to-end can indeed improve performance. Without carefully handcrafted dynamics and rendering models, this pipeline can reconstruct complex 3D dynamic information from 2.5D input.

The learned dynamics can be used to perform editing. And it is also impressive to see this method can also render the dynamic shadow correctly (as in the video demo).

**Weaknesses:**

The method is actually only capturing the surface points. There are no inner points reconstructed with only the depth values. We can also see this effect in the demo videos, where objects seem to be hollow.

The algorithm might rely on high-quality and multi-view RGBD videos. In the experiments, the background is clean and the objects are relatively simple. There are not so many tests on real-world data where the depth values are noisy and the view angles are sparse.

More like a systematic integration, this paper has relatively limited technical contribution. The feature extraction, neural renderer, and simulator use existing building blocks. It might be better if there more explorations on how to improve its interpretability, robustness, generalizability, and performance.

**Questions:**

What's the memory and time consumption to train this pipeline? The neural simulator might need to take many samples to converge while the NeRF rendering is usually slow (although there are faster implementations, I'm not sure if it's used here).

In Figure 8, while the PSNR stays the same between 1024->2048 and 4096->8192 while takes big jumps between 2048->4096 and 8192->16384.

This method seems to treat the same points in different views as different points. "If more than one camera is present, we can simply merge all the particle sets from each time-step"
Is this strategy appropriate, with unbalanced camera distribution? It would make more sense to me if additional steps could be taken to find and resolve the correspondence among views. And how is this method tacking the points across timesteps?

---

> ### Author Response · Authors · 2023-11-22
> **Reply 1 / 2**
>
> We thank the reviewer for their time and their detailed comments.
>
> Before diving into responses to the reviewer's questions, we would like to clarify an important point about how points are tracked across timesteps. In fact, VPD differs from the previous MeshGraphNets (MGN) [3] work in that it requires no tracking at all. This is a core contribution of this work: removing the need for tracking so that the method can plausibly scale to real-world data. MGN relies on ground truth tracking / correspondences through time in two main ways, and VPD introduces new methods that bypass that requirement:
>
> 1. Construction of inputs: MGN uses correspondence to compute velocity features for each particle, allowing the graph network to perceive motion at every location. VPD introduces a new graph structure where the particles for multiple input timesteps are included as different types of nodes in the graph network. VPD's hierarchical GNN then implicitly infers motion from this multi-timestep graph in order to make predictions.
> 2. Supervision: MGN uses correspondence to calculate point-wise L2 losses between a particle's predicted location and its true location at the next timestep. These correspondences are not available with RGB-D data, as objects move across the camera's field of view. Instead, VPD supervision is purely from next-timestep image prediction. The dynamics of the particles are supervised by the end-to-end gradients of this pixel-wise loss from the renderer. The structure of the renderer, which uses only features from nearby particles when rendering a specific location in space, means that the dynamics model learns to move particles in a way that follows the motion of objects. The dynamics model and renderer are designed to lead to this behavior, which is emergent and does not require any point tracking.
>
> These are fairly fundamental changes to the way MGN works in contrast to a simple integration of existing building blocks.
>
> > The method is actually only capturing the surface points.
>
> This is exactly right: VPD only represents a scene by points on the surface of the objects. As a result, it will be difficult to handle the dynamics of fluids or granular materials, at least without leveraging methods like [1, 2] which can fill volumes. However, surface points form a parsimonious representation of a scene that can capture many dynamics of interest without the O(n^3) cost of representing volumes.
>
> One natural concern with a surface representation like this is about what happens under occlusion: do missing points result in gaps in the 3D representation? We find that both the VPD dynamics model and renderer are able to handle the partial observability resulting from (self-)occlusion. For the MuJoCo Block dataset, conditioning on only one view gives comparable dynamics predictions to conditioning on 9 views (Fig. 6b), and we can see that the renderer is able to complete the back side of the block when rendering from novel views: https://sites.google.com/view/latent-dynamics#h.iz4y0q9qa9h3.
>
> > We can also see this effect in the demo videos, where objects seem to be hollow.
>
> We think the reviewer might be referring to the videos of the deformable objects. In that case, these objects are actually hollow, consisting of a thin rubbery material. The other datasets show solid objects, as shown by the dynamics, though they are still represented by points on the surface.

---

> ### Author Response · Authors · 2023-11-22
> **Reply 2 / 2**
>
> > The algorithm might rely on high-quality and multi-view RGBD videos. In the experiments, the background is clean and the objects are relatively simple. There are not so many tests on real-world data where the depth values are noisy and the view angles are sparse.
>
> Our experiments are in a simplified setting, but we find that in this setting VPD is robust to having few views and using depth predicted by a convnet instead of ground truth. Fig. 6b shows the performance of the model trained with varying numbers of conditioning views, and Fig. 6c shows that it performs nearly as well with predicted depth as with ground truth. The Deformables datasets have only four camera views, which is at least feasible to implement in the real world.
>
> Applying VPD to real-world data is a high priority piece of future work for us, though we were unable to do so for this submission due to a lack of pre-existing datasets with multi-view RGB-D data.
>
> > What's the memory and time consumption to train this pipeline?
>
> This model is memory-intensive, especially for the graph network, as the graphs contain ~30,000 points and ~100,000 edges. During training, with 6 recursive rollout steps and a backward pass, memory consumption is above 30GB, and training the whole model end to end takes 2-3 days. For training, the NeRF rendering speed is not a major bottleneck, since the model only renders a random subset of pixels for each training step, but it could be improved by skipping rendering in regions with no nearby points. Improving the efficiency of the model will be a key direction for future work.
>
> > In Figure 8, while the PSNR stays the same between 1024->2048 and 4096->8192 while takes big jumps between 2048->4096 and 8192→16384.
>
> Thanks for pointing this out. We have re-run this evaluation with more samples and updated Figure 8 in the paper.
>
> > This method seems to treat the same points in different views as different points.
>
> It would be interesting to investigate methods for explicitly merging and de-duplicating points which are very near to one another. So far this happens only implicitly in the model, which learns to cope with the non-uniform point sampling distribution. Note that this affects the set of inputs to the model but not the loss, which is computed on pixels.
>
> [1] Guan, Shanyan, et al. "Neurofluid: Fluid dynamics grounding with particle-driven neural radiance fields." International Conference on Machine Learning. PMLR, 2022.
>
> [2] Xue, Haotian, et al. "3D-IntPhys: Towards More Generalized 3D-grounded Visual Intuitive Physics under Challenging Scenes." Proceedings of the IEEE/CVF Conference on Computer Vision and Pattern Recognition. 2023.
>
> [3] Pfaff, T., Fortunato, M., Sanchez-Gonzalez, A., & Battaglia, P. W. (2020). Learning mesh-based simulation with graph networks. International Conference on Learning Representations.

---

### Author Response · Authors · 2023-11-22
**Overall response**

We thank all of the reviewers for the time and their constructive feedback. We appreciate their recognition of the technical soundness and significance of our work, along with its interesting editing capabilities and correct handling of dynamic shadows.

We have updated the text of the paper in response to the reviewers' feedback, and in addition performed several new experiments:
- We tested the ability of the model to generalize to larger scenes. We loaded a model trained on scenes with two objects, then composed much larger scenes of 32 objects, which we then simulated and rendered. We find that the dynamics remain well-behaved. Videos of this experiment are available on the video site: https://sites.google.com/view/latent-dynamics#h.sckus2cmnoec
- We created a new dataset with a mix of deformable and rigid objects and trained a single VPD model to predict their dynamics. This learned multi-material simulator is able to use visual features (color) to implicitly determine which objects have which behavior and simulate them accordingly. Videos of this model are here: https://sites.google.com/view/latent-dynamics#h.13u38bw637iq
- We investigated the model's ability to reconstruct occluded parts of a scene. When conditioned on only one view, the model is still able to reconstruct the occluded back side of a block and the floor behind it: https://sites.google.com/view/latent-dynamics#h.iz4y0q9qa9h3

Due to the limited time available during this discussion period, these new experiments have only preliminary results. If the paper is accepted, we will add more details of these experiments to the camera-ready version of the paper.

Thanks again to all of the reviewers. We feel that their suggestions have improved this work.

---

### Meta-Review · Area_Chair_VPAE · 2023-12-07

**Metareview:**

The submission proposes to learn particle based simulators directly from observations, where no access to privileged information is necessary, in particular no correspondences. The paper has received favorable reviews from 4 experts, who appreciated:

- an interesting research direction and a clear motivation,
- good performance,
- an interesting choice of representation, which allows a wide choice of downstream applications.

While a limited methodological contributions was criticised, the reviewers (and AC) consider that the paper is a valuable contribution to the field. The AC recommends acceptance.

**Justification For Why Not Higher Score:**

It could go higher, but since the methodological contribution is somewhat limited, I do not suggest it.

**Justification For Why Not Lower Score:**

There is no reason to reject this paper

---

### Decision · Program_Chairs · 2024-01-16

Accept (poster)